# Applying Proteomics and Computational Approaches to Identify Novel Targets in Blast-Associated Post-Traumatic Epilepsy

**DOI:** 10.3390/ijms25052880

**Published:** 2024-03-01

**Authors:** Jack L. Browning, Kelsey A. Wilson, Oleksii Shandra, Xiaoran Wei, Dzenis Mahmutovic, Biswajit Maharathi, Stefanie Robel, Pamela J. VandeVord, Michelle L. Olsen

**Affiliations:** 1School of Neuroscience, Virginia Polytechnic Institute and State University, Blacksburg, VA 24061, USA; jackb7@vt.edu; 2Genetics, Bioinformatics and Computational Biology, Virginia Polytechnic Institute and State University, Blacksburg, VA 24061, USA; 3Department of Biomedical Engineering and Mechanics, Virginia Polytechnic Institute and State University, Blacksburg, VA 24061, USA; kelseyaw@vt.edu; 4Department of Biomedical Engineering, Florida International University, Miami, FL 33174, USA; oshandra@fiu.edu; 5Virginia-Maryland College of Veterinary Medicine, Virginia Polytechnic Institute and State University, Blacksburg, VA 24061, USA; xrwei@vt.edu; 6Department of Cell Developmental and Integrative Biology, University of Alabama at Birmingham, Birmingham, AL 35294, USA; dzenis@vt.edu (D.M.); srobel@uab.edu (S.R.); 7Neurology & Rehabilitation, University of Illinois, Chicago, IL 60612, USA; bmahar2@uic.edu; 8Salem Veteran Affairs Medical Center, Salem, VA 24153, USA

**Keywords:** traumatic brain injury, post-traumatic epilepsy, concussive brain injury, blast neurotrauma, gene ontology

## Abstract

Traumatic brain injury (TBI) can lead to post-traumatic epilepsy (PTE). Blast TBI (bTBI) found in Veterans presents with several complications, including cognitive and behavioral disturbances and PTE; however, the underlying mechanisms that drive the long-term sequelae are not well understood. Using an unbiased proteomics approach in a mouse model of repeated bTBI (rbTBI), this study addresses this gap in the knowledge. After rbTBI, mice were monitored using continuous, uninterrupted video-EEG for up to four months. Following this period, we collected cortex and hippocampus tissues from three groups of mice: those with post-traumatic epilepsy (PTE^+^), those without epilepsy (PTE^−^), and the control group (sham). Hundreds of differentially expressed proteins were identified in the cortex and hippocampus of PTE^+^ and PTE^−^ relative to sham. Focusing on protein pathways unique to PTE^+^, pathways related to mitochondrial function, post-translational modifications, and transport were disrupted. Computational metabolic modeling using dysregulated protein expression predicted mitochondrial proton pump dysregulation, suggesting electron transport chain dysregulation in the epileptic tissue relative to PTE^−^. Finally, data mining enabled the identification of several novel and previously validated TBI and epilepsy biomarkers in our data set, many of which were found to already be targeted by drugs in various phases of clinical testing. These findings highlight novel proteins and protein pathways that may drive the chronic PTE sequelae following rbTBI.

## 1. Introduction

The Centers for Disease Control (CDC) report that epilepsy affects 3.4 million, or 1.2%, of the US population [1]. Epilepsy is defined as two or more spontaneous or unprovoked recurrent seizures. Though many brain insults may lead to seizures, traumatic brain injury (TBI) is a leading cause of acquired epilepsy, accounting for 10–20% of reported cases [2]. Acquired epilepsy, associated with TBI, or post-traumatic epilepsy (PTE) demonstrate a poor response to current antiepileptic drugs (AEDs) [3], necessitating mechanistic studies identifying disrupted proteins and protein pathways for the development of effective therapeutic strategies.

It has been reported that more than 25% of Veterans suffered from closed-head TBI due to blast wave exposure produced by explosive devices or high caliber weapons during combat, and most have been exposed to multiple low-level blasts within their tour of duty [4]. Assigned a unique medical code in 2023, in the CDC *International Classification of Diseases*, 10th Revision, Clinical Modification (ICD-10-CM), rbTBI is described to be associated with acute cognitive and behavioral deficits, with a subset of those injured having persistent debilitating effects including headaches, anxiety, depression, and sleep disturbances [5,6]. Blast TBI is of significant concern, particularly to the military population, wherein it has been dubbed the ‘signature wound’ of Operations Enduring Freedom, Iraqi Freedom, and New Dawn (OEF/OIF/OND) [7]. A recent study investigating the neurological sequelae of repeated low-level blast exposure in a population of Special Operations Forces services members utilized PET-CT imaging to identify markers of brain inflammation, which supported previous work demonstrating changes in cortical thickness, white matter, and default-mode network connectivity in military and law enforcement personnel exposed to repeated low-level blast exposure [8]. Additionally, 3 out of 16 blast-exposed OEF/OIF Veterans were clinically diagnosed with PTE [9]. Moreover, a diagnosis of non-convulsive seizures was suspected in 44% of the group. These and other limited studies have prompted the Department of Veterans Affairs to highlight epilepsy as a major concern for the long-term care of Veterans [9,10] and emphasize the need to account for the underlying molecular mechanisms that may serve to drive PTE following blast injury.

One significant difficulty in evaluating PTE is the highly variable time delay between the TBI and the onset of epilepsy, which, in humans, may manifest as long as a decade after injury [11,12], making unequivocal associations between epilepsy and TBI challenging. Epidemiological studies suggest that the severity of the injury correlates with an increased likelihood of PTE and that those patients who suffer early seizures (i.e., within seven days of the injury) are more likely to progressively develop PTE [11]. This delay in PTE onset suggests a potential therapeutic window for prevention and has been the study of intensive investigations using preclinical models. In vitro and in vivo models of single bTBI and repeated bTBI (rbTBI) have been developed. In rodent models of bTBI, studies have indicated sleep disturbances, anxiety-like behavior, and elevated conditioned fear [13,14,15]. Experimental evidence indicates markers of inflammation, early (<48 h) non-convulsive seizures, and neuronal hyperexcitability, as well as long-term spontaneous seizure development in 46% of animals [16], causally linking rbTBI to PTE. However, the underlying cellular and molecular mechanisms that drive these changes are unknown.

Here, we present an unbiased proteomics analysis at a chronic timepoint (four months) in a mouse model of rbTBI, with the goal of identifying proteins and protein pathways that contribute to the rbTBI-associated chronic sequalae, including PTE. At four months post injury, cortex and hippocampal tissue from sham, PTE^+^, and PTE^−^ were subject to shotgun LC–MS/MS. Using a combined experimental and computational approach, evidence is presented that identifies several PTE^+^ protein targets linked to mitochondrial dysfunction that could be considered as potential targets for drug intervention.

## 2. Results

### 2.1. Repeated Blast TBI Caused Spontaneous, Unprovoked, Recurrent Seizures

To establish seizures and PTE in rbTBI we implemented chronic, continuous (24/7) monitoring of brain activity using video-EEG. Considering the low incidence of seizures reported in rodent models of post-traumatic epilepsy, [17,18] mice were equipped with EEG electrodes 24 h after experiencing blast TBI and continuously recorded from two days post injury (dpi) and up to 106 dpi (Figure 1A). Overall, 11 out of 53 (21%) rbTBI mice developed spontaneous, unprovoked, recurrent seizures. Prior to and/or following the ictal episode, intermittent epileptiform inter-ictal activity was observed, characterized by paroxysmal spike discharges (sharp-shaped potentials lasting 30–70 milliseconds) and spike/sharp wave discharges (SWDs) with durations under 200 milliseconds (Figure 1C). Seizures were generalized as having been evidenced by the presence of epileptic activity across all EEG channels. It is worth noting that, after most seizures, a suppression in the EEG background, known as post-ictal suppression, was observed ranging from several seconds and up to several minutes (Figure 1C).

### 2.2. Chronic Protein Dysregulation in the Cortex and Hippocampus following rbTBI

An unbiased proteomics approach was used to identify proteins and protein pathways that were impacted as a result of rbTBI at four months post injury. Here, a subset of sham, rbTBI PTE^−^, and rbTBI PTE^+^ (five per group) mice from EEG studies were selected at random for unbiased LC–MS/MS whole cortical and hippocampal proteomics. In total, untargeted proteomics quantified the relative abundance of 3377 proteins through the identification of 22,620 unique peptide ions across all biological replicates in the hippocampus and cortex with a high FDR confidence level (FDR < 0.01) and peptide spectrum match > 3 (Appendix A). Following data processing, differentially expressed proteins (DEPs) were identified as having a fold change (FC) ≥ |1.2|, an adjusted *p*-value < 0.1, and presented in at least 5 out of 10 technical replicates in the overexpressing group. In total, 282, 273, and 225 DEPs were identified comparing PTE^−^ vs. sham, PTE^+^ vs. sham, and PTE^+^ vs. PTE^−^ in the cortex, and 323, 314, and 248 DEPs in PTE^−^ vs. sham, PTE^+^ vs. sham, and PTE^+^ vs. PTE^−^, in the hippocampus, respectively (Figure 2A–D). Venn diagrams indicate the number of shared or unique DEPs within each comparison for the cortex and hippocampus (Figure 2A,B). The top 10 DEPs in each group are highlighted in the volcano plots (Figure 2C,D). Protein abundance, fold change, and statistical data regarding groups can be found in Appendix A. Several shared DEPs were observed between sham vs. PTE^−^ and sham vs. PTE^+^, suggesting abundant general injury-induced changes. Along with shared DEPs, there were many that were unique to the PTE^+^ vs. PTE^−^ group, indicating seizure-related proteins. Of the shared and unique DEPS, these include upregulation of Fxyd1, Tbc1d15, and Trapp13 in the cortex and Mblac2 in the hippocampus (Figure 2C,D). Notably, the protein Cap2, cyclase-associated protein 2, an actin-binding protein, which is found in neuronal growth cones and associated with morphology of dendritic spines (for review, see [19]), was upregulated in the PTE^−^ vs. PTE^+^ comparison in both the cortex and hippocampus. Additional seizure- and epilepsy-associated proteins identified in PTE^+^ and PTE^−^ cortex and hippocampus include Scnb1 [20], Isca2 [21], Wdr4 [22], and Dkc1 [23]. In each comparison, several DEPs identified were below the level of detection in one of the comparing groups (PTE^−^ vs. sham, PTE^+^ vs. sham, and PTE^+^ vs. PTE^−^), including 31, 33, and 38 for each cortical comparison, and 41, 43, and 37 proteins in the hippocampal comparisons. These data are provided in Appendix A (0.1 = protein not detected in experimental group, 100 = protein not detected in control group). After identifying the top 10 dysregulated DEPs in each comparison per brain region, z-scores were computed from raw signal intensity for each biological replicate within the groups, where PTE^−^ and PTE^+^ samples appeared to cluster together in the cortex, in contrast to sham and PTE^−^ samples clustering together in the hippocampus (Figure 2E,F) (denoted by dotted boxes). These results were further confirmed by comparing the average signal intensities of the top 10 upregulated and top 10 downregulated DEPs per comparison for each sample in either the cortex or hippocampus in a correlation heatmap (nearly 60 proteins in total, Figure 2G,H). These findings indicate an injury-related effect in the rbTBI cortex, but a seizure-related effect in the rbTBI hippocampus.

### 2.3. Functional Enrichment Analysis of Dysregulated DEPs in rbTBI

Functional enrichment analysis was performed to identify overrepresented proteins that may serve to drive rbTBI-associated phenotypes, including seizures across the cortex and hippocampus of sham and rbTBI animals. Here, DEPs were separated by log_2_FC as upregulated and downregulated. From these lists, gene symbols were used to determine gene ontology (GO) assignments using the *cluster profiler* R package to identify over and under expressed GO pathways. The top 10 upregulated and downregulated biological processes (BPs) in each comparison were plotted utilizing the GOPlot package in R, based on all dysregulated DEPs. In the hippocampus of rbTBI mice, several biological process (BP) terms relating to synapse organization, neurogenesis, and the electron transport chain, as well as ATP synthesis and metabolism were identified (Figure 3A). While comparing sham vs. PTE^−^ and sham vs. PTE^+^, we identified similar enriched pathways in the cortical and hippocampal brain regions relating to mitochondrial dysfunction and oxidative phosphorylation (Appendix A). Intriguingly, many pathways in the PTE^+^ vs. PTE^−^ comparison were similar to the sham comparisons, suggesting heightened mitochondrial dysfunction in either the PTE^−^ or PTE^+^ group.

In the hippocampal PTE^+^ vs. PTE^−^ comparison, enrichment terms suggested alterations in mitochondrial function and cellular metabolism, as has been reported in blast injury in several recent studies [24,25,26,27]. To further investigate processes related to mitochondrial dysfunction, terms related to ATP synthesis and metabolism, aerobic respiration, and acyl co-A metabolism, as well as the synthesis and metabolism of purines and their highly integrated proteins were plotted using a chord plot (Figure 3B). The analysis revealed thar the protein Gmpr3, guanosine monophosphate reductase 2, an enzyme that catalyzes the irreversible and NADPH-dependent reductive de-amination of guanosine monophosphate (GMP) to inosine monophosphate (IMP), was the most highly upregulated protein. In contrast, Ndufv3, NADH:Ubiquinone oxidoreductase subunit V3, an accessory subunit of the mitochondrial membrane respiratory chain NADH dehydrogenase (Complex I), was the most downregulated protein in this analysis. Here also Gcdh, Npc1, Acly, Dip2a, and Mpc1, were highly integrated proteins, each being represented in seven of the ten enriched GO terms (Figure 3B). 

### 2.4. Predicted Alterations in Flux Reactions Caused by DEPs in PTE^+^ rbTBI

The GO analysis presented above predicts disrupted protein pathways but falls short of identifying their effects on the substrates and metabolites both consumed and produced from these pathways. To better understand the changes occurring to the metabolic network and cellular metabolism between the hippocampal PTE^+^ and PTE^−^ comparison, a system-level analysis was conducted using the application of a constraint-based reconstruction and analysis (COBRA) method known as flux balance analysis (FBA). We decided to perform FBA by integrating our data with the RECON3D model from the Biochemical, Genetic, and Genomic (BiGG) database. Briefly, this approach can be seen as an optimization problem that searches for sets of steady-state reaction fluxes that maximize or minimize an objective function representing a given biological purpose through linear programming. Through comparative analysis of the reaction fluxes solved by various objective functions (ATP maintenance, ATP synthesis, nutrient update), it was observed that biomass production accounted for a broader range of cellular activities and could reflect a more physiologically relevant objective compared to a singular focus. Thus, we chose biomass maintenance, which focuses on optimizing the distribution of metabolic fluxes of those reactions required to meet the energy and resource requirements necessary for maintaining cellular functions, as the objective function. This would allow us to initially evaluate the distribution of fluxes and to optimize the network to represent a normal, steady-state condition. Recognizing protein synthesis as an outcome of gene expression, we incorporated this aspect into our modeling approach. Specifically, values representing protein fold changes were linked to the bounds of reaction fluxes regulated by these proteins, originally provided by the optimized solution. Metabolite outputs for each reaction between the steady-state and disease condition were then evaluated.

When the objective function in FBA was set to biomass maintenance, the reaction CYOO3mi (#1, Table 1) (Cytochrome c Oxidase Complex IV, CYOOm3i) flux was 3.187 mol hr-1gDW-1 and reaction CYOR_u10mi (#2, Table 1) (Ubiquinol-10 cytochrome c reductase, Complex III) flux was 1.813 mol hr-1gDW-1, for proton production in the inner mitochondrial membrane. In the disease condition, these flux values decreased to 0.917 mol hr-1gDW-1 and 1.152 mol hr-1gDW-1, respectively, indicating a 58% reduction in proton production. Additional dysregulated flux reactions observed include SUCD1m (#3, Table 1) (Succinate Dehydrogenase), AKGDm (#5, Table 1) (2-oxoglutarate dehydrogenase), ECOAH9m (#4, Table 1) (Enoyl-coa hydratase, 3-Hydroxy-2-methylbutyryl-CoA forming), GMPR (#6, Table 1) (GMP Reductase), and PGK (#7, Table 1) (Phosphoglycerate kinase). Additional detailed descriptions of these reactions can be found in the BiGG database [28]. These results highlight a nuanced metabolic shift, characterized by heightened metabolic activity and an increased demand for these reactions, while simultaneously predicting a reduction in proton production across the inner mitochondrial membrane. 

### 2.5. Identification of High Confidence Targets for Drug Intervention in Seizure Prevention

In the last set of predictive approaches, we used unbiased proteomic data to find potential drug targets for treating or preventing seizures. We started by analyzing the high confidence protein interactions of the DEPs through the search tool for the retrieval and of interacting genes/proteins (STRING) database. A high confidence interaction is defined by a confidence threshold exceeding 0.7. Subsequently, we created a protein-protein interaction network (PPIN) based on these identified interactions where each “node” is represented by a protein, and each “edge” is represented as the interaction or association between them. This network was then imported into Cytoscape and centrality measurements were computed to quantitatively assess the significance of individual nodes in the network. To concentrate our efforts on interventions that are more likely to influence the overall network dynamics, the key nodes identified may serve as suitable drug targets. In our analysis, we prioritized degree and stress centrality as the most crucial measurements giving these a high weight, emphasizing nodes with a high number of direct connections and those acting as key bottlenecks in the network’s shortest path. Betweenness, closeness, and eccentricity metrics were also weighted in descending order of importance to enhance our analysis by identifying nodes that act as critical bridges, efficient network communicators, and hold a unique positional significance. The top 10 nodes with the highest weighted sums and an FC greater than |1.5| were identified as Cpeb2, Echs1, Hnrnpk, Mbn11, Nduf4b, Ndufv3, Pten, Ube2d3, and Uqcrb (refer to Figure 4 and Table 2, protein function provided). Additional proteins, namely Atp5pf, Kng1, Krt1, Nrp1, and Vps41 (Table 3), were identified through a second form of weighted analysis, PageRank. All 15 central nodes identified have been experimentally validated as dysregulated in either epilepsy or TBI (Table 4 and Table 5). Moreover, four of these central nodes are present in publicly available biomarker datasets, and eight have been inferred with a high score to contribute to each condition, further reinforcing their status as high-confidence targets (see Table 4). The remaining nodes represent potential avenues for future study. 

To characterize the druggability of the identified central nodes, we utilized all 10 of the nodes identified in our centrality analysis and the additional 5 identified by PageRank. Here, the DrugBank (https://go.drugbank.com/), Pharos (https://pharos.nih.gov/), and Drug-Gene (https://www.dgidb.org/) interaction databases were mined to identify drugs which have been created to target these nodes, ranging from experimental to approved clinical use, in mice and humans. All of the identified nodes have identified drugs which targeted them (Table 5). These findings suggest the possibility for the repurposing of these drugs to be utilized in the treatment of rbTBI-associated epilepsy. 

Most of the central proteins identified are notably understudied in the context of TBI-PTE. Thus, we further interrogated small molecules, biologics, and other therapeutic modalities using drug target ontology (DTO) families to target the 15 central node proteins and their predicted post-translation modifications (Table 5). Using Pharos, the target development level (TDL) [57] of each central protein was identified with the majority belonging to the category of ‘Tbio.’ This category encompasses proteins that have gene ontology (GO) leaf term annotations supported by experimental evidence, or that meet at least two out of three of the following conditions: a fractional publication count exceeding five, three or more Gene RIF annotations, or 50 or more commercial antibodies, as counted in the Antibodypedia portal. The remaining central proteins fell into either the ‘Tchem’ category, confirming their ability to interact with small molecules with a high potency, or the ‘Tclin’ category, validating their association with approved drugs. Two central proteins, however, fall into the fourth TDL category ‘Tdark,’ which includes 31% of human proteins curated at the primary sequence level in UniProt, but do not belong to an additional category. In addition to the TDL, their DTO family was also identified to further investigate if those central proteins would make for a suitable drug target, given our data highlighting their critical role in TBI-PTE. Most of the central proteins identified belonged to the DTO families of transporter proteins. Of note, transporter proteins, as well as enzymes and nuclear receptors, are the main classes of proteins that are considered a part of the druggable proteome.

## 3. Discussion

In the current study, an unbiased proteomics screening was performed in sham, rbTBI PTE^+^, and rbTBI PTE^−^ mice. Here, we report spontaneous recurrent seizures in a subset of animals exposed to rbTBI (21%), as has been previously reported [16,58]. Given the limited number of preclinical molecular studies using rbTBI models to assess chronic timepoints, a discovery-based, unbiased approach was performed to identify the molecular drivers of rbTBI-related neurological dysfunction. Using shotgun LC–MS/MS, hundreds of dysregulated proteins were observed in the cortex and hippocampus of sham vs. rbTBI mice, with the results largely finding differences in protein expression between rbTBI PTE^+^ animals vs. seizure-free mice (PTE^−^). Here, the analysis revealed a significant protein pathway dysregulation in mitochondrial function, post-translational modifications, and protein transport. Applying computational approaches provided additional evidence of mitochondrial dysfunction, specifically in oxidative phosphorylation, and predicting distinct metabolic shifts occurring in the inner mitochondrial membrane in rbTBI PPIN and PageRank analysis performed on the proteomics data enabled the identification of 15 high-confidence protein targets that may serve as therapeutic targets in rbTBI PTE^+^, of which all have been experimentally validated as dysregulated in either epilepsy or TBI. Moreover, four of the central nodes we identified were also identified in publicly available biomarker datasets. Thus, the findings highlight the disrupted protein pathways that may serve to drive a chronic pathological function post rbTBI. Additionally, these datasets may serve as a useful resource for those interested in rbTBI-driven chronic pathological function and protein dysregulation post rbTBI.

PTE is an increasing concern in both civilian and military populations that are exposed to blast waves from an explosive device or high-caliber weapons, yet few studies have investigated blast-related PTE [9,10]. Bugay et al. (2020) [16] reported on the occurrence of PTE following repeated blast events in mice. Their study utilized EEG to identify seizures at one month following rbTBI, finding the development of PTE in 46% of animals. While the results of this current study reported a lower incidence of PTE following rbTBI, differences in the injury device and blast protocol used could account for the discrepancy. Furthermore, it should be noted that, following TBI in humans, PTE may manifest as far as a decade post TBI [11,12], thus, had we followed our animals for longer, we may have observed higher seizure incidence. The rbTBI model presented here resulted in spontaneous recurrent seizures, which began acutely and continued through four months. This model may, thus, serve to study the chronic effects of blast related injury and to aid the development of novel clinical treatment strategies.

Mitochondrial dysfunction has previously been reported as a result of bTBI. In mouse models of blast injury, investigations at early time points (6 h) reveal decreased neuronal ATP levels, which rebound by 24 h [59]. Additional acute effects post rbTBI include decreased mitochondrial membrane potential, increased release of cytochrome C, and upregulation of Caspase-3, indicative of early apoptosis [60]. Further, Hubbard et al. reported mitochondrial dysfunction in synaptic mitochondria across the brain, with elevated levels of glial oxidative stress 48 h post rbTBI [24], and oxidative phosphorylation and mitochondrial dysfunction were observed in a mouse model of rbTBI at both the 7 and 30 days post injury timepoints [27]. Additional literature links rbTBI to alterations in cerebrovascular structure [61,62,63] and the neurovascular unit, including swollen astrocyte perivascular endfeet with disorganized organelles, as early as 24 h post injury [64] and a complete loss of perivascular endfeet at chronic timepoints [62]. Our studies examined the whole cortical and hippocampal tissue rather than discrete cell populations, thus we are unable to conclude if the mitochondrial effects we observed are associated with unique cell populations. However, based on the above-mentioned findings related to cerebrovascular remodeling post rbTBI, there exists the possibility that deficits in cerebrovascular function drive neural cell shifts in metabolic demand. Collectively, this accumulating body of literature utilizing different rbTBI paradigms and different model systems (rat and mouse) supports acute and chronic alterations in tissue metabolism and mitochondrial dysfunction, possibly driven by altered cerebral blood perfusion, a condition which may be exacerbated by PTE.

Numerous methods have been developed to measure and analyze biological systems across various omics platforms, yet, systematically assessing metabolic systems remains challenging, as gene expression or protein levels may not always directly correlate with metabolic activity. To provide a more comprehensive understanding on how the dysregulated proteins may serve to alter the metabolic network and the cellular metabolism associated with PTE, this study applied a systematic analysis using FBA, a COBRA technique. The COBRA framework utilizes a stoichiometric matrix to convert mass-balanced metabolic reactions within a cellular system, encompassing both uptake and secretion rates, into a comprehensive matrix that delineates alterations in the levels of reactants and products for each reaction. The choice of FBA permitted the use of the RECON3D model, the most extensive genome-scale model of human metabolism, incorporating data on 3288 open reading frames responsible for encoding metabolic enzymes that catalyze 13,543 reactions involving 4140 distinct metabolites [65]. Our unbiased approach predicts dysregulated flux across complexes I-IV of the electron transport chain (ETC), including a reduction in proton production across the inner mitochondrial membrane. Of these complexes, complex IV, which transfers electrons from cytochrome C to oxygen and adds to the maintenance of the proton gradient, was found to have the highest metabolite dysregulation. This proton gradient is critical for the generation of ATP through oxidative phosphorylation, ultimately suggesting dysregulated cellular metabolism in the hippocampus of PTE+ rbTBI. Protein pump inhibitors, such as pantoprazole, have been associated with an increased risk of developing seizures, particularly in the elderly population [66]. Notably, it should be recognized that there are limitations in the FBA analysis, including the utilization of protein fold changes to introduce constraints in the FBA model and of proteins dysregulated in the PTE^+^ condition, which were absent in the Recon3D model, each of which may reduce the accuracy of the FBA model. Yet, despite these limitations, the results provide compelling evidence of mitochondrial dysfunction, specifically in oxidative phosphorylation, and distinct metabolic shifts occurring in the inner mitochondrial membrane. While we cannot definitively conclude that these changes are a result of seizures, our results support a recent and growing body of literature that mitochondrial dysfunction contributes to persistent neurological dysfunction in rbTBI.

Proteomic approaches represent a powerful tool for advancing biomarker discovery and therapeutic target identification. While not as sensitive as transcriptional profiling, an understanding of dysregulated proteins and their functional pathways provides a unique insight into drug and biomarker discovery [67]. Here, by integrating the proteomics data set with computational protein–protein interaction networks, 15 high-confidence targets for drug intervention in seizure prevention were identified. Remarkably, all of the 15 identified targets have been associated with the epilepsy and/or TBI literature (Table 4). These results highlight the utility of an unbiased approach and indicate that rbTBI within the context of PTE shares common molecular signatures with other forms of TBI. Future analysis on the PTE proteome will be required to determine if rbTBI alone is sufficient to induce a similar molecular signature. Additionally, further investigation into Tdark-identified proteins, proteins which have not been extensively annotated, may provide a more comprehensive view of the molecular events associated with TBI-PTE. These proteins may also uncover previously known aspects of the disease’s pathophysiology and might exhibit individual variability, allowing for a more targeted and effective therapeutic intervention. Although several promising anti-epileptogenic drugs and disease modifying therapies have been tested in preclinical and clinical trials, they are far from satisfactory in preventing the chronic effects from blasts, such as PTE. Therefore, there is a need for novel or alternative treatment strategies for rbTBI to prevent such outcomes. To this end, the high-confidence targets were classified using Pharos, to assess their suitability for drug development. Additionally, PTMs of the identified targets were predicted using MusiteDeep, as these modifications can provide insights into the molecular mechanisms, as well as gauge the feasibility of developing therapeutics against these targets. These findings warrant additional exploration into the influence of post-translational modifications on regulating the assembly and functionality of electron transport chain complexes in PTE. Available drugs targeting our high confidence targets were also identified, suggesting potential drug repurposing.

Together, our results indicate long-term changes in the hippocampal and cortical proteome following rbTBI, regardless of PTE status. Further, mitochondrial function was perturbed in both regions and was further dysregulated in PTE, as indicated by the presence of protein and pathways in the PTE^+^ vs. PTE^−^ comparison groups. These pivotal advancements that aligned experimental and computational approaches have the potential to develop, expand, and optimize more effective treatments for PTE.

## 4. Materials and Methods

### 4.1. Animal Experimentation

All experiments were conducted in accordance with the NIH Guide for the Care and Use of Laboratory Animals and with the approval of the Virginia Tech Institutional Animal Care and Use Committee. C57/Bl6 male and female mice at P60–90 from Charles River, were housed in an AAALAC-accredited facility with a 12 h light–dark cycle, food, and water, ad libitum. For EEG-video recording, animals were housed individually in a 12.5″ × 12.5″ × 15.5″ polycarbonate cage (AAA Plastic Products, Birmingham, AL, USA) with corncob bedding and nesting material [68]. An established preclinical model of blast neurotrauma was used for this study [15]. An advanced blast simulator (ABS) generated free-field blast waves [10]. Isoflurane anesthetized mice (4% for 5 min) prior to the blast were positioned inside the ABS in a mesh sling that allows for minimal hindrance of the blast wave through the simulator and prevents wall impact. Static overpressures were calculated to be 13.8 ± 0.433 psi at the animal’s position using the Rankine–Hugoniot equation. Repeated bTBI was induced with three blasts at one hour inter-injury intervals. Sham-injured mice underwent the entire process without blast injury.

### 4.2. EEG Electrode Placement and Video-EEG Data Acquisition

Stereotactic surgery was performed on all sham and rbTBI mice 24 h after blast, as previously described [69]. Briefly, mice were anesthetized with 3.5% isoflurane for 5 min before placement into the stereotactic apparatus (Kopf Instruments, Tujunga, CA, USA). Buprenorphine (0.1 mg/kg) was administered subcutaneously for analgesia. The isoflurane level was maintained at 1.5% during the surgery. Stainless steel electrodes (0.10″ screw with wire leads (#8403, Pinnacle Technology, Parsippany, NJ, USA) or 00-96 × 1/16 (1.6 mm) screws with wire leads (Plastics1)) were implanted epidurally for chronic intracranial recordings (Figure 1A). Wire leads from the electrodes were connected into the custom 6-pin connector for 2 monopolar and 1 bipolar EEG channels *(*#8235-SM-C, Pinnacle Technology) or 6 channel electrode pedestals for 3 monopolar EEG channels (#MS363, Plastics One, Roanoke, VA, USA). For EEG acquisition, we used two systems: Pinnacle Technology (8200-K1-SE system) and Biopac Systems (Goleta, CA, USA) (ERS100C amplifiers, MP160 system) at the 2000 Hz sampling rate with a high-pass cutoff at 1 Hz (Pinnacle Technology and Biopac Systems), and a low-pass cutoff at 1000 Hz (Pinnacle Technology) and 3000 Hz (Biopac Systems). For synchronized 24/7 video data acquisition, cameras had either integrated or separate infrared sources for dark hours video acquisition and were affixed over (dome cameras) or in front of (box cameras) the single-housed Plexiglas cylinders. Video-EEG data were collected continuously (24/7) with programmed, automated restart every 24 h for four months. EEG acquisition was acquired using Sirenia Acquisition v2.2.11 (Pinnacle Technology) and Acqknowledge 5 (Biopac Systems) software.

### 4.3. Seizure Detection

EEG for seizure detection and characterization data were converted to EDF format and were first analyzed manually. Subsequently, data were re-analyzed using a custom automated Matlab algorithm with post hoc manual proofreading and validation by experimenters blinded to the experimental group. The following criteria were used to identify electrographic seizures: event length of at least 5 s, an amplitude at least three times greater than the background signal, and the presence of an evolution of amplitude and frequency of the event.

### 4.4. Cortical and Hippocampal Dissection and Dissociation 

Four months post rbTBI, sham and injured mice were anesthetized using CO_2_, and were decapitated. The cortex and hippocampus were dissected from each hemisphere and were separated in ice-cold ACSF (120 mM NaCl, 3.0 mM KCl, 2 mM MgCl, 0.2 mM CaCl, 26.2 mM NaHCO_3_, 11.1 mM glucose, 5.0 mM HEPES, 3 mM AP5, 3 mM CNQX), bubbled with 95% oxygen. The right cortical and hippocampal hemisphere were separately minced into 1 mm^3^ pieces to form a homogeneous tissue. An aliquot of tissue was homogenized in protein homogenization buffer consisting of 50 mM Tris-HCl, 150mM NaCl, 1% Triton X-100, 0.5% Sodium deoxycholate, and 0.1% sodium dodecyl sulfate, and a Bradford assay was used to determine protein concentration (Thermo Scientific, Waltham, MA, USA, Cat #: 23225).

### 4.5. Trypsin Digestion and Untargeted LC–MS/MS Proteomic Acquisition

One hundred micrograms of protein from each animal/tissue region was adjusted to a final volume of 100 μL using 2X S-Trap protein solubilization buffer (10% (*w*/*v*) SDS, 200 mM triethylammonium bicarbonate (TEAB), pH 8.5). Each sample per group was run twice to obtain two technical replicates per sample. Samples were reduced and alkylated by incubation at 37 °C for 1 h with 4.5 mM dithiothreitol (DTT), followed by incubation at room temperature for 30 min in the dark with 10 mM iodoacetamide (IAA). Unreacted IAA was quenched by the addition of DTT to 10 mM. Protein was precipitated by the addition of 10 μL 12% (*v*/*v*) *o*-phosphoric acid and 1ml methanol and incubated overnight at −80 °C. Precipitated protein was pelleted using centrifugation at 13,000× *g* and 4 °C for 20 min and loaded onto S-Traps (Protifi, Fairport, NY, USA) at 1000× *g* and room temperature for 1 min. After extensive washing of the pellets entrapped in the S-Traps using methanol, protein was digested by incubation at 37 °C for 4 h after the addition of 1 μg trypsin in 50 mM TEAB, followed by a second addition of 1 μg trypsin in 50 mM TEAB and incubation at 37 °C overnight. Peptides were recovered from S-Traps after digestion by sequential washing with solvent A (2:98 acetonitrile:water supplemented with 0.1% (*v*/*v*) formic acid) and solvent B (80:20 acetonitrile:water supplemented with 0.1% (*v*/*v*) formic acid). After removal of acetonitrile using a centrifugal vacuum concentrator, peptide concentrations were determined by measuring the absorbance at 215 nm. Samples were diluted using solvent A to a concentration of 0.5 mg/mL based on these measurements. Duplicate injections of 10 μL (approximately 5 μg) of the resulting peptide mixture for each sample were analyzed utilizing our standard 2 h DDA LC–MS/MS method on a Thermo Orbitrap Fusion Lumos instrument (Thermo Fisher Scientific, Waltham, MA, USA).

### 4.6. Untargeted Proteomic Data Analysis

Raw data were processed using Thermo Proteome Discoverer 2.5 and searched using two different search engines (Mascot (Matrix Science, Columbus, OH, USA) and Sequest HT (Thermo)) using the mouse protein database from Uniprot, appended with a database containing common laboratory contaminants. Results from both search engines are reported together as one result file. Search parameters were limited to trypsin-specific peptides with two possible missed cleavages, precursor, and fragment mass tolerances of 10 ppm, a fixed modification of carbamidomethylation on Cys, and variable modifications of oxidation of Met, deamidation of Gln/Asn, and pyroGlu formation from Gln at the N-terminus of a peptide.

Peptide abundances were normalized so that the sum of the intensities for all identified peptides are equal in each sample run. Protein abundances were then calculated as the sums of the peak intensities for all peptides associated with each peak. Abundances for technical replicates are averaged prior to averaging across biological replicates. Abundance ratios for all relevant comparisons were given for each identified protein with a corresponding *p*-value calculated via *t*-tests using the Benjamini–Hochberg method to adjust for false discovery rates. Male mice were primarily utilized for proteomics, except for two female mice in the PTE^+^ group. Using the R packages ggfortify (v0.4.16) and ggplot2 (v3.4.3), generated 2D PCA plots were consistently clustered into sham, PTE^+^, and PTE^−^ animals, thus males and females were collapsed in each group (Appendix A). Identified unique peptides and proteins are provided in Appendix A and identification files provided in the ProteomeXchange with the accession number PXD048228. All differentially expressed proteins were then plotted utilizing ggplot2, tidyverse (v2.0.0), and ggrepel (v0.9.3) packages in R to generate volcano plots, along with the venn (v1.11) package for venn diagrams and lattice (v0.21.8), reshape2 (v1.4.4), and ggplot2 for heatmaps in R. All DEPs were, again, used for GO enrichment with the R package clusterProfiler (v4.4.4) and org.Mm.eg.db (v3.15.0). The top 10 most significantly upregulated and downregulated biological process (BP) terms were used to generate GO bubble plots with the R package ggplot2. Following GO enrichment, DEPs found in both the PTE^+^ and PTE^−^ groups of the hippocampus were plotted on a chord plot utilizing the GOplot (v1.0.2) package in R.

### 4.7. Methodological Approach for Predicting Cellular Metabolism

A system-level analysis was conducted using the application of a constraint-based reconstruction and analysis (COBRA) method known as flux balance analysis (FBA) [70]. FBA was performed by integrating identified proteomics DEP with the RECON3D model (http://bigg.ucsd.edu/models/Recon3D, accessed on 2 October 2023). Using Biomass maintenance as the objective function, values representing protein fold changes were linked to the bounds of reaction fluxes regulated by these proteins, originally provided by the optimized solution. Metabolite outputs and functional relations among fluxes between steady state and the disease condition were then evaluated.

### 4.8. Protein Network Centrality Analysis

Significant DEPs were entered into the STRING database to construct a comprehensive protein–protein interaction network (PPIN) and imported into Cytoscape. The construction of this network involved setting a minimum interaction score threshold (0.7) to ensure high confidence in the interactions. Subsequently, centrality measurements were computed within Cytoscape using the ‘Network Analyzer’ function. These centrality metrics were then exported and further analyzed in Python (3.11). In the Python analysis, the ‘pandas v1.24’ library was employed alongside ‘numpy v1.5.2’, a fundamental package for scientific computing. ‘Pandas’ facilitated the efficient handling and manipulation of tabular data, while ‘numpy’ provided essential support for mathematical operations and array manipulations. The proteins within the network were then ranked based on their weighted sums, where the weights were assigned according to the perceived importance of each centrality measure in the overall network. Degree and stress were given the highest weights, followed by betweenness, closeness, and eccentricity, in descending order. Additionally, for comparative analysis, the PageRank algorithm [71] was applied to evaluate the importance of a node, by considering both the number and quality of its connections within the network.

## Figures and Tables

**Figure 1 ijms-25-02880-f001:**
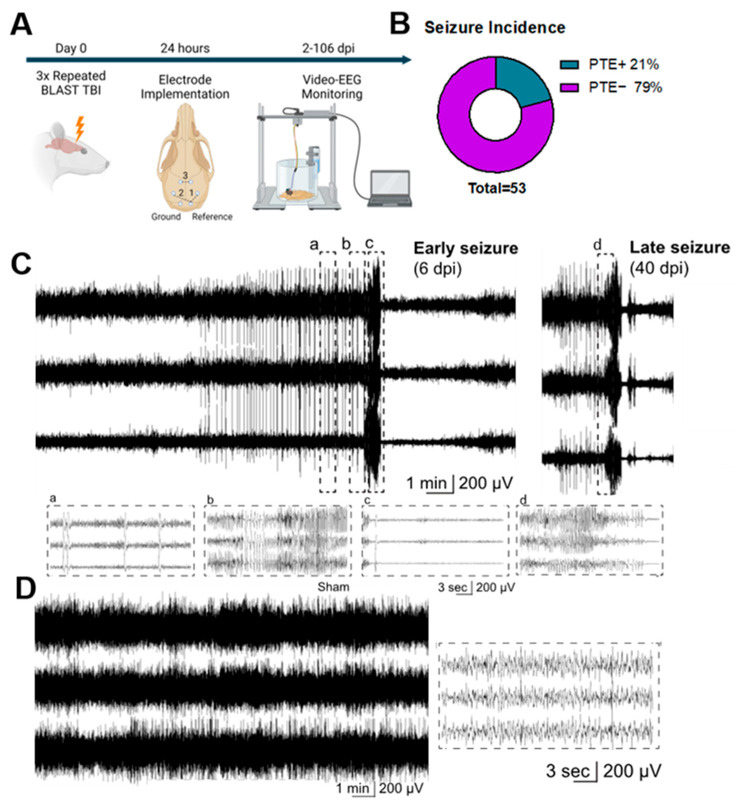
rbTBI induced spontaneous, unprovoked, recurrent seizures in mice. (**A**): EEG study design. Day 0—induction of BLAST TBI. Then, 24 h after TBI, mice underwent stereotactic surgery and electrode implantation. Mice were connected to the video-EEG acquisition system on post-TBI day 2 and were recorded up to 106 days. Diagram created with BioRender.com (**B**): Overall incidence of seizures. (**C**): Representative EEG traces of early (6 days post injury) and late (40 days post injury) electro-clinical seizures in rbTBI mouse. Panel a: typical pre-ictal and inter-ictal activity in the form of generalized bilateral epileptic spikes; panel b: expanded view of the active seizure phase; panel c: post-seizure depression; panel d: expanded view of the late seizure. (**D**): Representative EEG traces of physiological background activity of a sham mouse.

**Figure 2 ijms-25-02880-f002:**
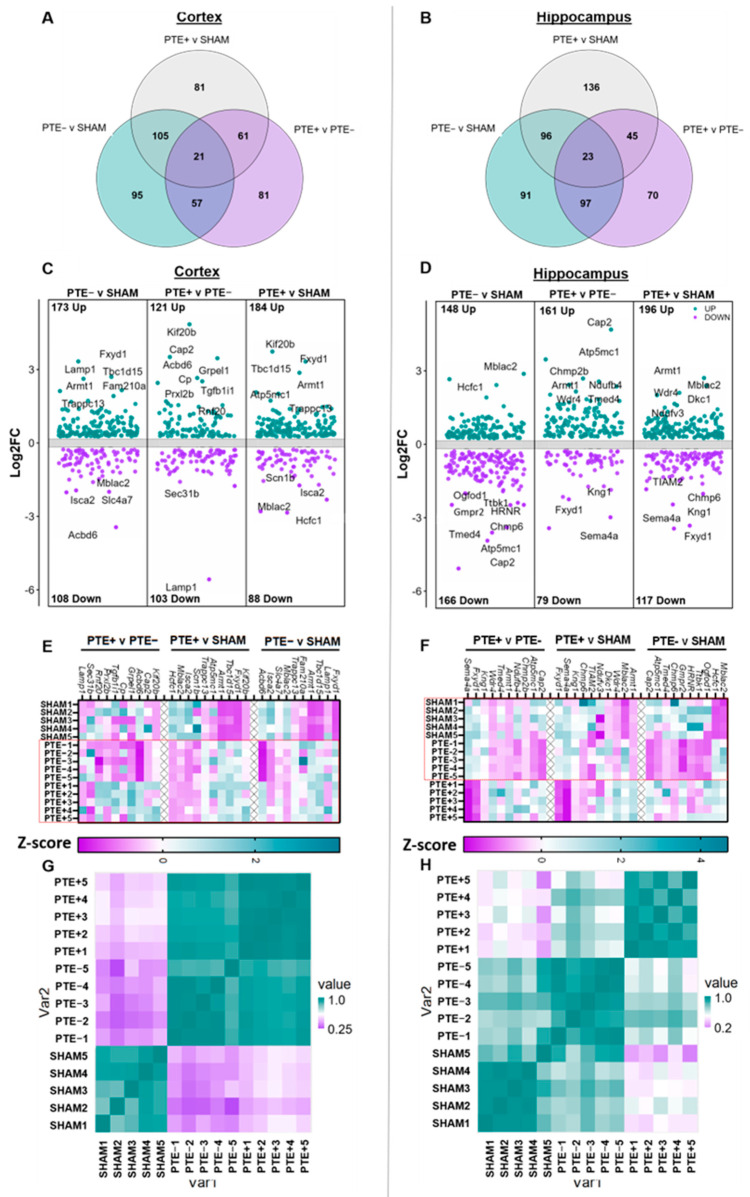
Identification of DEPs’ rbTBI-associated PTE^−^ and PTE^+^ cortex and hippocampus. (**A**,**B**) Venn diagrams showing unique and shared DEPs between each comparison in the cortex (left) and the hippocampus (right). (**C**,**D**) Volcano plots of all DEPs in each comparison from either the whole cortex (left) or the whole hippocampus (right) with an adjusted *p*-value ≤ 0.1 and an FC greater than l 1.2 l, with the top ten dysregulated proteins marked. (**E**,**F**) Heatmap of the top 10 dysregulated DEPs in each comparison in either the whole cortex (**left**) or the whole hippocampus (**right**) based on z-score. Dotted box in E marks injury related effects, while the dotted box in F marks a seizure related effect. (**G**,**H**) Correlation heatmaps from either whole cortex (**left**) or whole hippocampus (**right**) samples based on the top 10 upregulated and top 10 downregulated DEPs’ intensity value in each comparison (PTE^−^ vs. sham, PTE^+^ vs. sham, and PTE^+^ vs. PTE^−^, for a total of 60 proteins per brain region).

**Figure 3 ijms-25-02880-f003:**
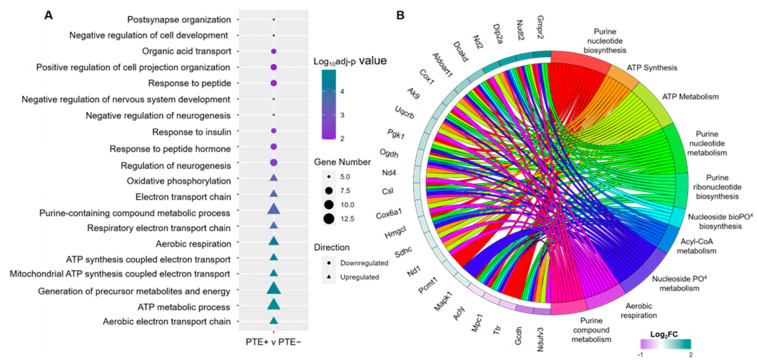
Functional enrichment analysis identifies disrupted biological processes related to the mitochondrial functional rbTBI-associated hippocampus. (**A**) Top 10 downregulated (circles) and top 10 upregulated (triangles) GO biological process pathways identified in the whole hippocampus PTE^+^ vs. PTE^−^ comparison. (**B**) GO chord plot indicating shorthand descriptions of biological processes relating to mitochondrial dysfunction and oxidative phosphorylation dysregulated in the whole hippocampus PTE^+^ vs. PTE^−^ comparison and the corresponding DEPs present in the gene list per term with associated log_2_FC.

**Figure 4 ijms-25-02880-f004:**
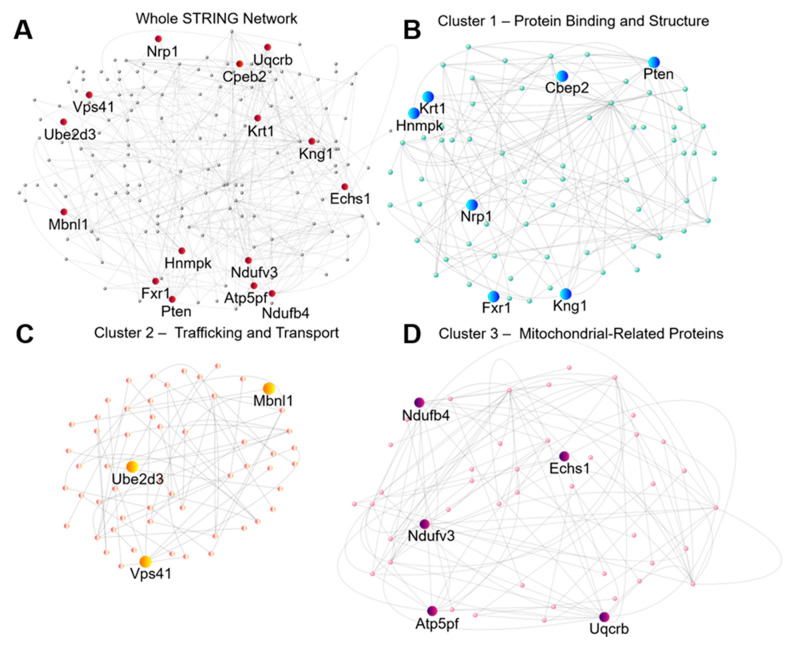
STRING analysis identifies three distinct sub-networks through k-means clustering. (**A**) Whole network of DEPs from the hippocampal PTE^+^ vs. PTE^−^ comparison. Nodes with the highest centrality through analysis or unbiased PageRank system are colored as red. (**B**) Sub-network denoting proteins related to protein binding and structural proteins, with high centrality or PageRank nodes marked in blue. (**C**). Sub-network denoting proteins related to trafficking and transport, with high centrality or PageRank nodes marked in orange. (**D**). Sub-network denoting mitochondrial proteins, with high centrality or PageRank nodes marked in purple.

**Table 1 ijms-25-02880-t001:** Optimized and Dysregulated Fluxes of Metabolites. Reaction IDs and corresponding flux for optimized and dysregulated states are listed above. Subscripts denote the location of the metabolite where m = mitochondria, I = inner mitochondrial membrane, and c = cytosol. Numbers in parentheses indicate the flux value for that metabolite as mol/hr^−1^/gDw^−1^. Negative values indicate metabolites consumed in the reaction, whereas positive values indicate metabolites being produced.

#	Reaction ID	Optimized Flux Equation (mol hr-1gDw-1)	Dysregulated Flux Equation (mol hr-1gDw-1)
1	CYOO3mi	7.92H_m_ (−3.187) + O2_m_ (0.7967) + 4FocytC_m_ (−3.187) ⇌ 1.96 H2O_m_ (1.561) + 4.0FicytC_m_ (3.187) + 0.02O2s_m_ (0.015) + 4H_i_ (3.187)	7.92H_m_ (−13.940) + O2_m_ (−1.761) + 4.0FocytC_m_ (−7.043) ⇌ +1.96H2O_m_ (3.451) + 4.0FicytC_m_ (7.043) + 0.02O2s_m_ (0.035) + 4.0H_i_ (0.917)
2	CYOR_u10mi	2.0H_m_ (−0.907) + 2.0FicytC_m_ (−0.907) +q10H2_m_ (−0.453) ⇌ 2.0FocytC_m_ (0.907) + q10_m_ (0.453) + 4.0H_i_ (1.813)	2.0H_m_ (−1.360) + 2.0FicytC_m_ (−1.360) + q10H2_m_ (-0.680) ⇌ 2.0FocytC_m_ (1.360) + q10_m_ (0.680) + 4.0H_i_ (1.152)
3	SUCD1m	FAD_m_ (−0.578) + Succ_m_ (−0.578) ⇌FADH2_m_ (0.578) + Fum_m_ (0.578)	FAD_m_ (−0.752) + Succ_m_ (−0.752) ⇌ FADH2_m_ (0.752) + Fum_m_ (0.752)
4	ECOAH9m	H2O_m_ (0.791) + 2mb2CoA_m_ (0.791) ⇌ 3hmbCoA_m_ (−0.791)	H2O_m_ (1.107) + 2mb2CoA_m_ (1.107) ⇌ 3hmbCoA_m_ (−1.107)
5	AKDGm	Akg_m_ (−0.366) + CoA_m_ (−0.366) + NAD_m_ (−0.366) ⇌ CO2_m_ (0.366) + NADH_m_ (0.366) + SucCoA_m_ (0.366)	Akg_m_ (−0.513) + CoA_m_ (−0.513) + NAD_m_ (−0.513) ⇌ CO2_m_ (0.513) + NADH_m_ (0.513) + SucCoA_m_ (0.513)
6	GMPR	GMP_c_ (0.027) + 2.0H_c_ (0.054) + NADPH_c_ (0.027) ⇌ imp_c_ (-0.027) + NADP_c_ (−0.027) + NH4_c_ (−0.027)	GMP_c_ (0.109) + 2.0H_c_ (0.218) + NADPH_c_ (0.109) ⇌ imp_c_ (−0.109) + NADP_c_ (−0.109) + NH4_c_ (−0.109)
7	PGK	3pg_c_ (0.449) + ATP_c_ (0.449) ⇌ 13dpg_c_ (−0.449) + ADP_c_ (−0.449)	3pg_c_ (0.674) + ATP_c_ (0.674) ⇌ 13dpg_c_ (−0.674) + ADP_c_ (−0.674)

**Table 2 ijms-25-02880-t002:** Hippocampal PTE^+^ vs. PTE^−^ protein network centrality analysis.

Gene Symbol	Degree ^#^	Betweenness Centrality ^$^	Closeness Centrality ^%^	Stress Centrality ^&^	Adj *p* Value	Protein Function *
*Cpeb2*	2	0.166667	0.292683	26	8.78 × 10^−2^	mRNA binding, regulating cytoplasmic polyadenylation of mRNA
*Echs1*	2	0.054054	0.355769	180	4.43 × 10^−6^	Catalyzes CoA intermediates to L-3-hydroxyacyl-CoAs in mitochondrial fatty acid beta-oxidation pathway
*Fxr1*	2	0.30303	0.375	48	2.67 × 10^−16^	RNA binding protein which shuttles between the nucleus and cytoplasm to bind to polyribosomes
*Hnrnpk*	4	0.712121	0.48	118	2.28 × 10^−6^	RNA binding protein which complexes with heterogeneous nuclear RNA and influence pre-mRNA processing and metabolism
*Mbnl1*	2	0.166667	0.352941	26	1.93 × 10^−2^	C3H-Zinc finger binding protein which modulates external splicing of pre-mRNAs
*Ndufb4*	18	0.001661	0.506849	32	2.67 × 10^−16^	Non-catalytic subunit of multisubunit NADH:oxidoreductase
*Ndufv3*	15	0.000374	0.474359	8	2.55 × 10^−6^	Subunit of multisubunit NADH:oxidoreductase; function unknown
*Pten*	3	0.7	0.625	14	1.17 × 10^−3^	Tumor suppressor which negatively regulates AKT/PKB signaling. Longer isoform may play a role in energy metabolism in the mitochondria
*Ube2d3*	3	0.166667	0.315789	38	3.04 × 10^−2^	Member of the E2 ubiquitin conjugating enzyme family, which functions in ubiquitination tumor suppressor protein p53
*Uqcrb*	19	0.006285	0.569231	114	2.07 × 10^−3^	Binds ubiquinone and participates in electron transfer while bound to ubiquinone

^#^ Interactions (edges) a protein has in the network. High degree centrality are hub proteins well connected within the network. Typically play a role in numerous biological pathways so targeting these can disrupt various processes. ^$^ Quantifies how often a protein acts as a bridge, connecting other proteins in the network. A high degree of betweenness is central to mediating the flow of information between different parts of the network. Targeting these could disrupt the flow of information and control over various processes. ^%^ Measures how close a protein is to all other proteins in the network, identifying proteins that efficiently interact with others. Targeting these may lead to widespread network perturbation and the modulation of multiple pathways. ^&^ Identifies proteins’ importance based on its role in connecting other nodes in a network. High stress centrality nodes serve as critical intermediaries in the network and are often seen as bridges or bottlenecks. * Protein functions obtained from GeneCards.

**Table 3 ijms-25-02880-t003:** Top 10 Nodes weighted through PageRank.

Gene Symbol	Protein Function *
*Atp5pf*	F6 subunit of the F0 complex, required for F1 and F0 interactions
*Hnrnpk ^#^*	RNA binding protein which complexes with heterogeneous nuclear RNA and influence pre-mRNA processing and metabolism
*Kng1*	Uses alternative splicing to generate high and low molecular weight kininogens
*Krt1*	Member of the keratin family, which are expressed during simple and stratified epithelial cell differentiation
*Ndufb4 ^#^*	Non-catalytic subunit of multisubunit NADH:oxidoreductase
*Nrp1*	Cell surface receptor involved in the development of the cardiovascular system, in angiogenesis, in the formation of certain neuronal circuits
*Pten ^#^*	Tumor supressor which negatively regulates AKT/PKB signaling. Longer isoform may play a role in energy metabolism in the mitochondria
*Ube2d3 ^#^*	Member of the E2 ubiquitin conjugating enzyme family, which functions in ubiquitination tumor suppressor protein p53
*Uqcrb ^#^*	Binds ubiquinone and participates in electron transfer while bound to ubiquinone
*Vps41*	Plays a role in transport and fusion of vacuoles from the Golgi

*^#^* Indicates nodes also detected through manual centrality analysis. * Protein functions obtained from GeneCards.

**Table 4 ijms-25-02880-t004:** Literature validation of PPIN and PageRank targets. Central proteins derived from proteomic analysis (adj *p*-value < 0.1, FC > |1.5|) of the hippocampus in mice with bTBI-associated PTE with selected literature support and clinical biomarker inclusion.

Gene Symbol	Supporting Epilepsy Literature	Supporting TBI Literature	Biomarker Validation Database					
			DisGenNet		* CTD		** Other	
*Cpeb2*	[29]	[29,30]	n	n	y	n	n	n
*Echs1*	[31]	[32]	n	n	y	n	y	n
*Fxr1*	[33]	[34]	n	n	y	y	n	n
*Hnrnpk*	[35]	[36]	n	n	y	n	n	n
*Mbnl1*	[37]	[38]	n	n	y	n	n	n
*Ndufb4*	[39]	[32]	n	n	y	n	n	n
*Ndufv3*	[39]	[40,41]	n	n	n	n	n	n
*Pten*	[39,42,43,44]	[45,46,47,48]	y	n	y	y	y	n
*Ube2d3*		[49]	n	n	y	n	n	n
*Uqcrb*	[50,51]	[32,52]	n	n		n	n	n
Additional nodes weighted through PageRank								
*Atp5pf*	[53]	[32]	n	N	y	n	n	n
*Kng1*	[54]	[32]	y	N	y	n	n	n
*Krt1*		[55]	n	N	n	n	n	n
*Nrp1*	[39]		n	N	y	n	y	n
*Vps41*	[56]	[56]	n	N	y	n	n	n

For DisGenNet, CTD, and other columns: left sub columns = epilepsy, right sub columns = TBI, Y = yes, N = no. (* CTD_Epilepsy inference score > 15) (CTD_TBI inference score > 3) (** include the databases OMIM, MGI, Pubchem, Uniprot, HGMD, and gene cards).

**Table 5 ijms-25-02880-t005:** Identified selective drugs and chemicals that bind or target the high-confidence putative markers in the hippocampus of mice with blast TBI-associated PTE.

Gene Symbol	Drug/Chemical	Drug/Chemical, Group	Database	Species		Protein Details		
				HU	MO	pPTMs	TDL	DTO Fam
*Cpeb2*	Bisphenol A	Experimental	CTD	y	y	p, me, ac, gl, ub, hy	Tbio	Nab
*Echs1*	Hexanoyl-CoA	Experimental	DrugBank	y	y	p, gl, ub, ac,	Tbio	E
*Fxr1*	Valproic Acid	Experimental	CTD	y	y	p, ac, pc, su, ub, gl, me, pa	Tbio	Nab
*Hnrnpk*	Phenethyl isothiocyanate, Artenimol	Investigational, Approved	DrugBank	y	y	gl, ub, ac, p, pc, hy, me, su	Tbio	E
*Mbnl1*	Bisphenol A	Experimental	CTD	y	y	pc, gl, p, ub, ac, hy, pa	Tbio	Nab
*Ndufb4*	Metformin	Approved	Pharos	y	y	me, su, p, gl, ub, ac	Tclin	E
*Ndufv3*	Metformin	Approved	Pharos	y	y	p, ub, gl, hy, su, ac	Tclin	na
*Pten*	Bisphenol A	Experimental	CTD	y	y	ac, ub, me, su, gl	Tbio	E
*Ube2d3*	Bisphenol A	Experimental	CTD	y	y	me, p, gl	Tchem	E
*Uqcrb*	Azoxystrobin	Experimental	DrugBank	y	y	ac, ub, p	Tbio	E
Additional nodes weighted through PageRank								
*Atp5pf*	Bisphenol A	Experimental	CTD	y	y	p, ub, ac, pc, su	Tbio	T
*Kng1*	Copper	Approved	DrugBank	y	y	pc, p, gl, ub, hy, me, ac	Tbio	EM
*Krt1*	Copper	Approved	DrugBank	y	y	p, pc, pa, me, gl, ac, ub, su	Tbio	E
*Nrp1*	PEGAPTANIB	Approved	CTD	y	y	su, p, ub, ac, gl, me, pc, hy,	Tchem	na
*Vps41*	Valproic Acid	Experimental	CTD	y	y	pc, p, su, gl, ub, ac, hy,	Tbio	T

Y = yes; N = no; HU = human; MO = mouse Target Development Level (TDL): T = Transporter; EM = Enzyme Modulator; na = not applicable; K = kinase; E = enzyme; Nab = nucleic acid binding. Post-Translational Modification Prediction (pPTM)—Phosphorylation: P, Glycosylation: gl, Ubiquitination: ub, SUMOylation: su, Acetylation: ac, Methylation: me, Pyrrolidone carboxylic acid: pc, Palmitoylation: pa, Hydroxylation: hy.

## Data Availability

The raw data supporting the conclusions of this article will be made available by the authors on request.

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
