# Peer review of "Applying Proteomics and Computational Approaches to Identify Novel Targets in Blast-Associated Post-Traumatic Epilepsy"

_ijms, 2024, doi:10.3390/ijms25052880_

Round 1

Reviewer 1 Report

Comments and Suggestions for Authors

This a well written manuscript that focuses to identify molecular changes in the blast associate post traumatic epilepsy. This also provides important insight about the potential druggable targets to treat post-traumatic epilepsy.

Minor comments:

1. Why was p value<1 used instead of <0.05 in Fig 2?

2. Is there any correlation between the severity of epileptic seizures and the extent of specific protein changes observed among PTE+ mice?

Reviewer 2 Report

Comments and Suggestions for Authors

The manuscript titled "Applying Proteomics and Computational Approaches to Identify Novel Targets in Blast Associated Post-Traumatic Epilepsy" hypothesizes that post-traumatic epilepsy (PTE) can result from traumatic brain injury (TBI), especially in Veterans with blast related TBI (bTBI). While the authors have conducted commendable research, specific points require consideration before advancing to the next stage. The following are points and suggestions for your consideration:

1.       Provide more details on the selection criteria for the mouse model of repeated blast TBI (rbTBI)? Why were this specific model chosen, and how does it mirror the conditions observed in Veterans with blast related TBI?

2.       How confident are authors in the accuracy and reproducibility of the reported protein expression changes?

3.       Are there any unexpected findings that warrant further investigation?

4.       How robust and reliable are the computational metabolic modeling predictions? Were these predictions validated or cross-verified through any independent experimental methods?

5.       Are there specific challenges or considerations in transitioning from preclinical findings to clinical trials?

6.       How might further investigation into 'Tdark' proteins contribute to our understanding of TBI-PTE?

7.       Given the heterogeneity of TBI and PTE, did you observe any variability in results among individual mice within the PTE+ and PTE- groups?

8.       What are the next steps or potential future directions for research based on the insights gained from this study?

9.       How might these findings inform subsequent experiments or clinical studies?

10.   Discuss any fundamental limitations in the mouse model of repeated blast TBI (rbTBI) used in this study and how well it captures the complexity of blast-related traumatic brain injuries observed in human Veterans?

11.   Considering the points raised, I recommend minor revisions to address these considerations and strengthen the manuscript further.

Reviewer 3 Report

Comments and Suggestions for Authors

I commend the authors for presenting a groundbreaking research article. The study employs an integrated approach, combining unbiased proteomics, continuous EEG monitoring, and computational modeling to unravel the molecular mechanisms of post-traumatic epilepsy (PTE) following repeated blast traumatic brain injury (rbTBI). The use of computational metabolic modeling, specifically flux balance analysis, enhances the study by predicting alterations in the metabolic network and cellular metabolism, offering a comprehensive understanding of the observed protein dysregulations. The identification of high-confidence drug targets through protein-protein interaction networks and drug databases is a notable strength, with potential implications for therapeutic strategies in rbTBI-associated epilepsy.

I have a couple of comments:

 1. Line 83: What is meant by "the first unbiased proteomics analysis"? Please provide clarification.

2. Lines 83-84: Why was 4 months post-TBI chosen as the endpoint of observation?

 3. Could you explicitly state the hypothesis and aim of this study?

 4. While the study observes mice for up to four months post-injury, it is essential to acknowledge that PTE in humans may manifest over a more extended period. The chosen timeframe might not capture the full spectrum of chronic sequelae.

 5. The introduction of computational metabolic modeling introduces complexity. The accuracy of predictions depends on assumptions and parameters in the model. It is crucial to recognize that these models might oversimplify or overlook certain aspects of the biological system.

Round 2

Reviewer 2 Report

Comments and Suggestions for Authors

The author answered all the queries raised by the reviewer, and the manuscript is improved. So it is now ready for publication.